# CARLA: A Python Library to Benchmark Algorithmic Recourse and Counterfactual Explanation Algorithms

**Martin Pawelczyk**[*]
University of Tübingen
martin.pawelczyk@uni-tuebingen.de

**Sascha Bielawski**
University of Tübingen
sascha.bielawski@uni-tuebingen.de

**Johannes van den Heuvel**
University of Tübingen
johannes.van-den-heuvel@uni-tuebingen.de

**Tobias Richter** [†]
CarePay International
t.richter@carepay.com

**Gjergji Kasneci** [†]
University of Tübingen
gjergji.kasneci@uni-tuebingen.de

## Abstract

Counterfactual explanations provide means for prescriptive model explanations by suggesting actionable feature changes (e.g., increase income) that allow individuals to achieve favourable outcomes in the future (e.g., insurance approval). Choosing an appropriate method is a crucial aspect for meaningful counterfactual explanations. As documented in recent reviews, there exists a quickly growing literature with available methods. Yet, in the absence of widely available open–source implementations, the decision in favour of certain models is primarily based on what is readily available. Going forward – to guarantee meaningful comparisons across explanation methods – we present CARLA (**C**ounterfactual **A**nd **R**ecourse **L**ibr**A**ry), a python library for benchmarking counterfactual explanation methods across both different data sets and different machine learning models. In summary, our work provides the following contributions: (i) an extensive benchmark of 11 popular counterfactual explanation methods, (ii) a benchmarking framework for research on future counterfactual explanation methods, and (iii) a standardized set of integrated evaluation measures and data sets for transparent and extensive comparisons of these methods. We have open sourced CARLA and our experimental results on Github, making them available as competitive baselines. We welcome contributions from other research groups and practitioners.

## 1 Introduction

Machine learning (ML) methods have found their way into numerous everyday applications and have become an indispensable asset in various sensitive domains, like disease diagnostics [13], criminal justice [4], or credit risk scoring [29]. While ML models bear the great potential to provide effective support in human decision making processes, their predictions may have considerable impact on personal lives, where the final decisions might be disadvantageous for an end user. For example, the rejection of a loan or the denial of parole might have negative effects on the future development of the corresponding person's life.

---

[*]Corresponding author
[†]Equal senior author contribution

35th Conference on Neural Information Processing Systems (NeurIPS 2021) Track on Datasets and Benchmarks.

When ML systems involve humans in the loop, it is crucial to build a strong foundation for long-term acceptance of these methods. To this end, it is critical (1) to *explain* the predictions of a model and (2) to *offer constructive means for the improvement* of those predictions to the advantage of the end–user. Counterfactual explanations – popularized by the seminal work of [59] – provide means for prescriptive model explanations by suggesting actionable feature changes (e.g., increase income) that allow individuals to achieve favourable outcomes in the future (e.g., insurance approval).

When counterfactual explainability is employed in systems that involve humans in the loop, the community refers to it as *recourse*. Algorithmic recourse subsumes precise recipes on how to obtain desirable outcomes after being subjected to an automated decision, emphasizing feasibility constraints that have to be taken into account. Those explanations are found by making the smallest possible change to an input vector to influence the prediction of a pretrained classifier in a positive way; for example, from 'loan denial' to 'loan approval', subject to the constraint that an individual's `sex` may not change. As documented in recent reviews, there exists a quickly growing literature with available methods (see Figure 1 and [53, 24, 58]), reflecting the insight that the understanding of complex machine learning models is an elementary ingredient for a wide and safe technology adoption.

In practice, the counterfactual explanation (CE) that an individual receives crucially depends on the method that computes the recourse suggestions. Hence, there is a substantial need for a standardized benchmarking platform, which ensures that methods can be compared in a transparent and meaningful way. Researchers need to be able to easily evaluate their proposed methods against the overwhelming diversity of already available methods and practitioners need to make sure that they are using the right recourse mechanism for the problem at hand. Therefore, a standardized framework for comparison and quality assurance is an essential and indispensable prerequisite.

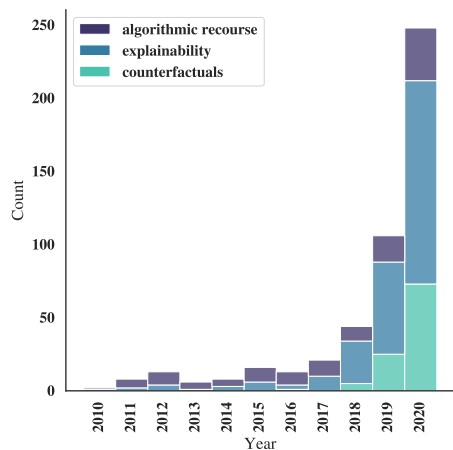

Figure 1: ArXiv submissions over time on explainability, counterfactual explanations and algorithmic recourse.

In this work, we present `CARLA` (**C**ounterfactual **A**nd **R**ecourse **L**ibr**A**ry), a python library with the following merits: First, `CARLA` provides *competitive baselines* for researchers to benchmark *new* counterfactual explanation and recourse methods for the standardized and transparent comparison of CE methods on different integrated data sets. Second, `CARLA` is a *common framework* with more than 10 counterfactual explanation methods in combination with the possibility to easily integrate new methods into a commonly accessible and easily distributable Python library. Moreover, the built-in integrated evaluation measures allow users to plug-in their custom black-box predictive models into the available counterfactual explanation methods and conduct extensive evaluations in comparison with other recourse mechanisms across different data sets. The same is true for researchers, who can use `CARLA` to extensively benchmark available counterfactual methods on popular data sets across various ML models. Third, `CARLA` *supports popular optimization frameworks* such as Tensorflow [1] and PyTorch [42], and provides a generic abstraction layer to support custom implementations. Users can can define problem–specific data set characteristics like immutable features and explicitly specify hyperparameters for the chosen counterfactual explanation method.

The remainder of this work is structured as follows: Section 2 presents related work, Section 3 formally introduces the recourse problem, Section 4 presents the benchmarking process. In Section 5, we describe our main findings, before concluding in Section 6. Appendices A - E describe `CARLA`'s software architecture and usage instructions, as well as additional experimental results, used ML classifiers, data sets and hyperparameters settings.

## 2   Related Work

Explainable machine learning is concerned with the problem of providing explanations for complex ML models. Towards this goal, various streams of research follow different explainability paradigms which can be categorized into the following groups [17, 14].

## 2.1 Feature Highlighting Explanations

**Local input attribution techniques** seek to explain the behaviour of ML models instance by instance. Those methods aim to understand how all inputs available to the model are being used to arrive at a certain prediction. Some popular approaches for model explanations aim at explainability by design [34, 2, 5, 60]. For white-box models – the internal model parameters are known – gradient-based approaches, e.g. [27, 6] (for deep neural networks), and rule-based or probabilistic approaches for tree ensembles, e.g. [19, 9] have been proposed. In cases where the parameters of the complex models cannot be accessed, model-agnostic approaches can prove useful. This group of approaches seeks to explain a model's behavior locally by applying surrogate models [49, 35, 50, 36], which are interpretable by design and are used to explain individual predictions of black-box ML models.

## 2.2 Counterfactual Explanations

The main purpose of counterfactual explanations is to suggest constructive interventions to the input of a complex model so that the output changes to the advantage of an end user. By emphasizing both the feature importance and the recommendation aspect, counterfactual explanation methods can be further divided into three different groups: independence-based, dependence-based, and causality-based approaches.

In the class of **independence-based methods**, where the input features of the predictive model are assumed to be independent, some approaches use combinatorial solvers or evolutionary algorithms to generate recourse in the presence of feasibility constraints [56, 51, 48, 23, 28, 8]. Notable exceptions from this line of work are proposed by [55, 32, 31, 18, 15], who use decision trees, random search, support vector machines (SVM) and information networks that are aligned with the recourse objective. Another line of research deploys gradient-based optimization to find low-cost counterfactual explanations in the presence of feasibility and diversity constraints [10, 38, 39, 52, 57, 45]. The main problem with these approaches is that they abstract from input correlations. That implies that the intervention costs (i.e., the costs of changing the input to achieve the proposed counterfactual state) are too optimistically estimated. In other words, the estimated costs do not reflect the true costs that an individual would need to incur in practical scenarios, where feature dependencies are usually present: e.g., *income* is dependent on *tenure*, and if *income* changes, *tenure* also changes (see Figure 2 for a schematic comparison).

In the class of **causality-based** approaches, all methods make use of Pearl's causal modelling framework [46]. As such, they usually require knowledge of the system of causal structural equations [20, 16, 25, 41] or the causal graph [26]. The authors of [25] show that these models can generate minimum-cost recourse, if the access to the true causal data generating process was available. However, in practical scenarios, the guarantee for such minimum-cost recommendations is vacuous, since, in complex settings, the causal model is likely to be miss-specified [26]. Since these methods usually require the *true* causal graph – which is the limiting factor in practice – we have not considered them at this point, but we plan to do that in the future.

**Dependence-based methods** bridge the gap between the strong independence assumption and the strong causal assumption. This class of models builds recourse suggestions on generative models [43, 11, 20, 37, 44]. The main idea is to change the geometry of the intervention space to a lower dimensional latent space, which encodes different factors of variation while capturing input dependencies. To this end, these methods primarily use variational autoencoders (VAE) [30, 40]. In particular, Mahajan et al. [37] demonstrate how to encode various feasibility constraints into VAE-based models. Most recently, [3] proposed CLUE, a generative recourse model that takes a classifier's uncertainty into account. Work that deviates from this line of research was done by [47, 22]. The authors of [47] provide FACE, which uses a shortest path algorithm on graphs to find counterfactual explanations. In contrast, Kanamori et al. [22] use integer programming techniques to account for input dependencies.

## 3 Preliminaries

In this Section, we review the algorithmic recourse problem and draw a distinction between two observational (i.e., non–causal) methods.

## 3.1 Counterfactual Explanations for Independent Inputs

Let $\mathcal{D}$ be the data set consisting of $N$ input data points, $\mathcal{D} = \{(x_1, y_1), \ldots, (x_N, y_N)\}$. We denote by $f : \mathbb{R}^d \to [0, 1]$ the fixed classifier for which recourse is to be determined. We denote the set of

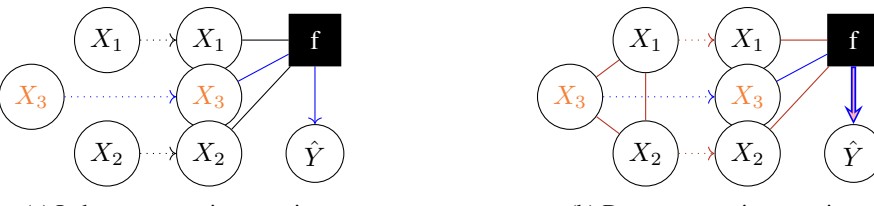

Figure 2: Different views on recourse generation. In (a) a change to $X_3$ only impacts $f$ through $X_3$, while in (b) the same change induces indirect effects on $f$, if $X_3$ is correlated with other inputs.

outcomes by $y(x) \in \{0,1\}$, where $y=1$ indicates the desirable outcome. Moreover, $\hat{y} = \mathbb{I}[f(x) > \theta]$ is the predicted class, where $\mathbb{I}[\cdot]$ denotes the indicator function and $\theta$ is a threshold (e.g., 0.5). Our goal is to find a set of actionable changes in order to improve the outcomes of instances $x$, which are assigned an undesirable prediction under $f$. Moreover, one typically defines a distance measure in input space $c : \mathbb{R}^d \times \mathbb{R}^d \to \mathbb{R}_+$. We discuss typical choices for $c$ in Section 4.

Assuming inputs are pairwise statistically independent, the recourse problem is defined as follows:

$$\delta_x^* = \arg\min_{\delta_x \in \mathcal{A}_d} c(x, \check{x}) \text{ s.t. } \check{x} = x + \delta_x, f(x + \delta_x) > \theta, \tag{I}$$

where $\mathcal{A}_d$ is the set of admissible changes made to the factual input $x$. For example, $\mathcal{A}_d$ could specify that no changes to sensitive attributes such as age or sex may be made. For example, using the independent input assumption, existing approaches [56] use mixed-integer linear programming to find counterfactual explanations. In the next paragraph, we present a problem formulation that relaxes the strong independence assumption by introducing generative models.

## 3.2 Recourse for Correlated Inputs

We assume the factual input $x \in \mathcal{X} = \mathbb{R}^d$ is generated by a generative model $g$ such that:

$$x = g(z),$$

where $z \in \mathcal{Z} = \mathbb{R}^k$ are latent codes. We denote the counterfactual explanation in an input space by $\check{x} = x + \delta_x$. Thus, we have $\check{x} = x + \delta_x = g(z + \delta_z)$. Assuming inputs are dependent, we can rewrite the recourse problem in (I) to faithfully capture those dependencies using the generative model $g$:

$$\delta_z^* = \arg\min_{\delta_z \in \mathcal{A}_k} c(x, \check{x}) \text{ s.t. } \check{x} = g(z + \delta_z), f(\check{x}) > \theta, \tag{D}$$

where $\mathcal{A}_k$ is the set of admissible changes in the $k$-dimensional latent space. For example, $\mathcal{A}_k$ would ensure that the counterfactual latent space lies within range of $z$. The problem in (D) is an abstraction from how the problem is usually solved in practice: most existing approaches first train a type of autoencoder model (e.g., a VAE), and then use the model's trained decoder as a deterministic function $g$ to find counterfactual explanations [20, 43, 37, 11, 3]. Our benchmarked explanation models roughly fit in one of these two categories.

## 4 Benchmarking Process

In this Section, we provide a brief explanation model overview and introduce a variety of explanation measures used to evaluate the quality of the generated counterfactual explanations. In Table 1 we present a concise explanation model overview.

### 4.1 Counterfactual Explanation Methods

AR (I) Ustun et al. [56] provide a method to generate minimal cost actions $\delta_x^*$ for linear classification models such as logistic regression models. AR requires the linear model's coefficients, and uses these coefficients for its search for counterfactual explanations. To provide reasonable actions it is possible to restrict $\delta_x^*$ to *user–specified constraints* (e.g., *has_phd* can only change from *False* to *True*) or to set a subset of inputs as *immutable* (e.g., *age*). The problem to find these changes is a discrete optimization problem. Given a set of actions, AR finds the action which minimizes a defined cost function, using integer programming solvers like CPLEX or CBC.

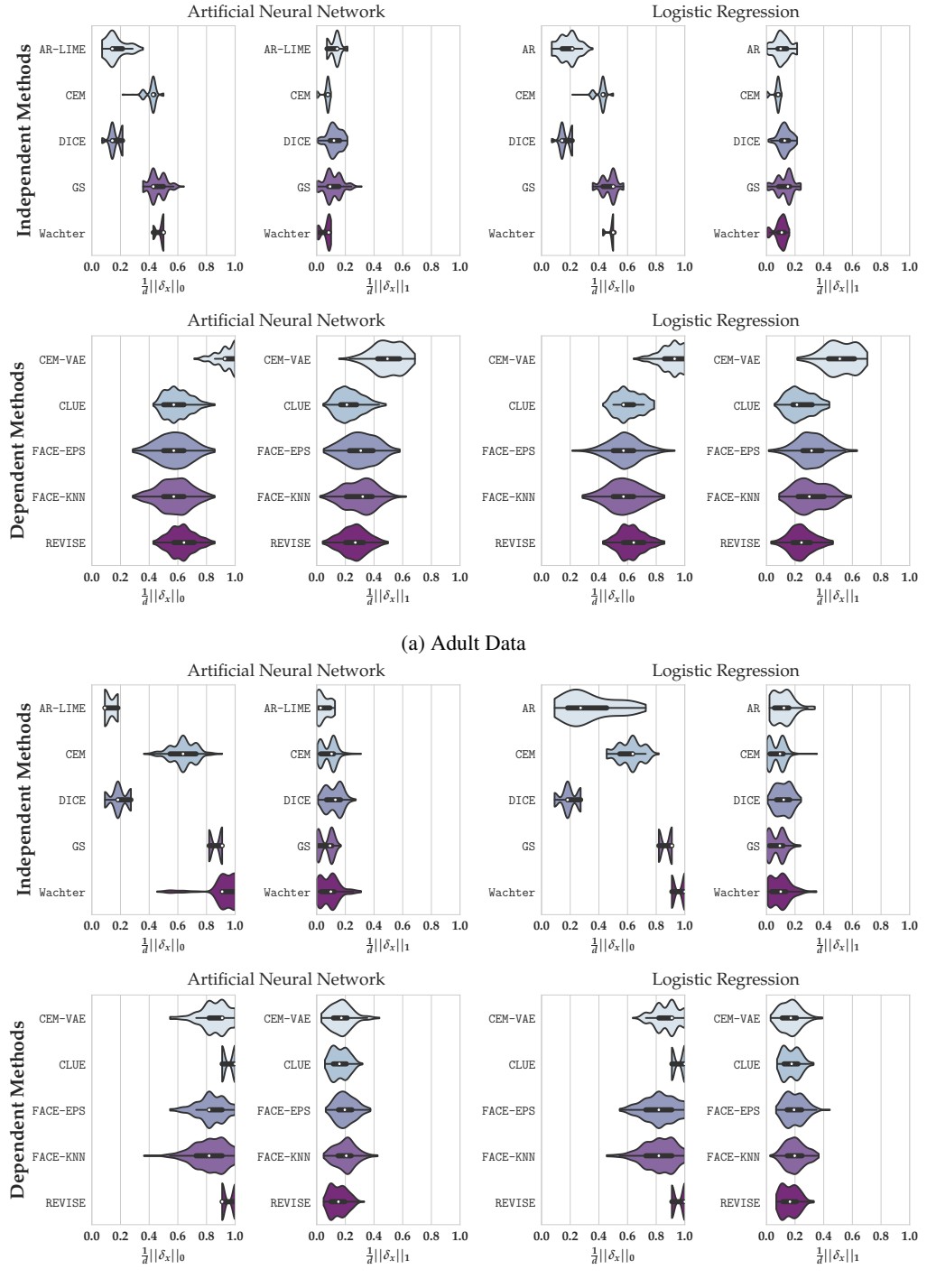

(a) Adult Data

(b) Give Me Some Credit Data

Figure 3: Evaluating the distribution of costs of counterfactual explanations on 2 different data sets (the results on COMPAS are relegated to Appendix B). For all instances with a negative prediction ($\{x \in \mathcal{D} : f(x) < \theta\}$), we plot the distribution of $\ell_0$ and $\ell_1$ costs of algorithmic recourse as defined in (1) for a logistic regression and an artificial neural network classifier. The white dots indicate the medians (lower is better), and the black boxes indicate the interquartile ranges. We distinguish between independence based and dependence based methods. The results are discussed in Section 5.

| Approach | Method | Model Type | Algorithm | Immutable | Categorical | Other |
|---|---|---|---|---|---|---|
| Independent (**I**) | AR | Linear | Integer Prog. | Yes | Binary | Direction of change |
| | AR-LIME | Agnostic | Integer Prog. | Yes | Binary | Direction of change |
| | CEM | Gradient based | Gradient based | No | No | None |
| | DICE | Gradient Based | Gradient based | Yes | Binary | Generative model |
| | GS | Agnostic | Random search | Yes | Binary | None |
| | Wachter | Gradient based | Gradient based | No | Binary | None |
| Dependent (**D**) | CEM-VAE | Gradient based | Gradient based | No | No | Gen. Model regularizer |
| | CLUE | Gradient based | Gradient based | No | No | Generative model |
| | FACE-EPS | Agnostic | Graph search | Binary | Binary | CE is from data set |
| | FACE-KNN | Agnositc | Graph search | Binary | Binary | CE is from data set |
| | REVISE | Gradient based | Gradient based | Binary | Binary | Generative model |

Table 1: Explanation method summary: we categorize different approaches based on their underlying assumptions and list what kind of ML model they work with (Model Type), the Method's underlying algorithm (Algorithm), whether the method can handle immutable features (Immutable), whether it can handle categorical features (Categorical) and any other outstanding characteristics (Other).

AR-LIME (**I**)    Most classification tasks do not have linearly separable classes and complex non–linear models usually provide more accurate predictions. Non–linear models are not per se interpretable and usually do not provide coefficients similar to linear models. We use a **reduction** to apply AR to non–linear models by computing a local linear approximation for the point of interest $x$, using LIME [49]. For an arbitrary black–box model $f$, LIME estimates post–hoc local explanations in form of a set of linear coefficients per instance. Using the coefficients we apply AR.

CEM (**I**)    Dhurandhar et al. [10] use an elastic–net regularization inspired objective to find low-cost counterfactual instances. Different weights can be assigned to $\ell_1$ and $\ell_2$ norms, respectively. There exists no immutable feature handling. However, we provide support for their VAE type regularizer, which should help ensure that counterfactual instances look more realistic.

CLUE (**D**)    Antorán et al. [3] propose CLUE, a generative recourse model that takes a classifier's uncertainty into account. This model suggests feasible counterfactual explanations that are likely to occur under the data distribution. The authors use a variational autoencoder (VAE) to estimate the generative model. Using the VAE's decoder, CLUE uses an objective that guides the search of CEs towards instances that have low uncertainty measured in terms of the classifier's entropy.

DICE (**I**)    Mothilal et al. [39] suggest DICE, which is an explanation model that seeks to generate minimum costs counterfactual explanations according to (**I**) subject to a diversity constraint which aims to promote a diverse set of counterfactual explanations. Diversity is achieved by using the whole range of suggested changes, while still keeping proximity to a given input. Regarding the optimization problem, DICE uses gradient descent to find a solution that trades-off proximity and diversity. Domain knowledge – in form of feature ranges or immutability constraints – can be added.

FACE (**D**)    The authors of [47] provide FACE, which uses a shortest path algorithm (for graphs) to find counterfactual explanations from high–density regions. Those explanations are actual data points from either the training or test set. Immutability constraints are enforced by removing incorrect neighbors from the graph. We implemented two variants of this model: the first variant uses an epsilon–graph (FACE-EPS), whereas the second variant uses a knn–graph (FACE-KNN).

Growing Spheres (GS) (**I**)    Growing Spheres – suggested in [32] – is a random search algorithm, which generates samples around the factual input point until a point with a corresponding counterfactual class label was found. The random samples are generated around $x$ using growing hyperspheres. For binary input dimensions, the method makes use of Bernoulli sampling. Immutable features are readily specified by excluding them from the search procedure.

REVISE (**D**)    Joshi et al. [20] propose a generative recourse model. This model suggests feasible counterfactual explanations that are likely to occur under the data distribution. The authors use a variational autoencoder (VAE) to estimate the generative model. Using the VAE's decoder, REVISE uses the latent space to search for CEs. No handling of immutable features exists.

Wachter (**I**)    The optimization approach suggested by Wachter et al. [59] generates counterfactual explanations by minimizing an objective function using gradient descent to find counterfactuals $\check{x}$ which are as close as possible to $x$. Closeness is measured in $\ell_1$-norm.

|  |  | Artificial Neural Network | | | | | Logistic Regression | | | | |
|---|---|---|---|---|---|---|---|---|---|---|---|
| Data Set | Method | *yNN* | redund. | violation | success | $\bar{t}(s)$ | *yNN* | redund. | violation | success | $\bar{t}(s)$ |
| | `AR(-LIME)` | 0.62 | **0.00** | 0.14 | 0.28 | 1.59 | **0.72** | 0.67 | 0.13 | 0.52 | 10.49 |
| | `CEM` | 0.26 | 3.96 | 0.66 | **1.00** | 1.10 | 0.20 | 3.98 | 0.66 | **1.00** | 0.92 |
| Adult | `DICE` | **0.71** | 0.53 | 0.17 | **1.00** | 0.13 | 0.58 | **0.51** | 0.23 | **1.00** | 0.13 |
| | `GS` | 0.30 | 3.77 | **0.09** | **1.00** | **0.01** | 0.30 | 3.94 | **0.10** | **1.00** | **0.01** |
| | `Wachter` | 0.23 | 4.45 | 0.83 | 0.50 | 15.72 | 0.16 | 1.67 | 0.94 | **1.00** | 0.03 |
| | `AR(-LIME)` | 0.89 | **0.00** | 0.29 | 0.07 | 0.55 | **1.00** | 2.33 | **0.14** | 0.39 | 3.42 |
| | `CEM` | **0.95** | 5.46 | 0.65 | **1.00** | 0.97 | 0.74 | 5.07 | 0.67 | **1.00** | 0.87 |
| GMC | `DICE` | 0.90 | 0.58 | 0.27 | **1.00** | 0.28 | 0.88 | **0.61** | 0.27 | **1.00** | 0.29 |
| | `GS` | 0.40 | 6.64 | **0.17** | **1.00** | **0.01** | 0.49 | 5.29 | 0.17 | **1.00** | **0.01** |
| | `Wachter` | 0.58 | 6.56 | 0.71 | **1.00** | 0.02 | 0.59 | 6.12 | 0.83 | **1.00** | **0.01** |

(a) Independence based methods

|  |  | Artificial Neural Network | | | | | Logistic Regression | | | | |
|---|---|---|---|---|---|---|---|---|---|---|---|
| Data Set | Method | *yNN* | redund. | violation | success | $\bar{t}(s)$ | *yNN* | redund. | violation | success | $\bar{t}(s)$ |
| | `CEM-VAE` | 0.12 | 9.68 | 1.82 | **1.00** | **0.93** | 0.43 | 10.05 | 1.80 | **1.00** | **0.81** |
| | `CLUE` | **0.82** | 8.05 | **1.28** | **1.00** | 2.70 | 0.33 | 7.30 | 1.33 | **1.00** | 2.56 |
| Adult | `FACE-EPS` | 0.65 | 5.19 | 1.45 | 0.99 | 4.36 | **0.64** | 5.11 | 1.44 | 0.94 | 4.35 |
| | `FACE-KNN` | 0.60 | **5.11** | 1.41 | **1.00** | 4.31 | 0.57 | **4.97** | 1.38 | 1.00 | 4.31 |
| | `REVISE` | 0.20 | 8.65 | 1.33 | 1.00 | 8.33 | 0.62 | 7.92 | **1.23** | **1.00** | 7.52 |
| | `CEM-VAE` | **1.00** | 8.40 | **0.66** | **1.00** | **0.87** | **1.00** | 8.54 | **0.36** | **1.00** | **0.88** |
| | `CLUE` | **1.00** | 9.39 | 0.90 | 0.93 | 1.91 | **1.00** | 9.56 | 0.96 | **1.00** | 1.76 |
| GMC | `FACE-EPS` | 0.99 | **8.06** | 0.99 | **1.00** | 19.44 | 0.98 | 7.98 | 0.96 | **1.00** | 19.50 |
| | `FACE-KNN` | 0.98 | 9.00 | 0.98 | **1.00** | 15.87 | 0.98 | **7.88** | 0.95 | **1.00** | 16.09 |
| | `REVISE` | **1.00** | 9.50 | 0.97 | **1.00** | 4.56 | **1.00** | 9.59 | 0.96 | 1.00 | 3.76 |

(b) Dependence based methods

Table 2: Summary of a subset of results for independence and dependence based methods. For all instances with a negative prediction ($\{x \in \mathcal{D} : f(x) < \theta\}$), we compute counterfactual explanations for which we then measure yNN (higher is better), redundancy (lower is better), violation (lower is better), success rate (higher is better) and time (lower is better). We distinguish between a logistic regression and an artificial neural network classifier. Detailed descriptions of these measures can be found in Section 4. The results are discussed in Section 5.

## 4.2   Evaluation Measures for Counterfactual Explanation Methods

As algorithmic recourse is a multi–modal problem we introduce a variety of measures to evaluate the methods' performances. We use six baseline evaluation measures. Besides distance measures it is important to consider measures that emphasize the *quality* of recourse.

**Costs**   When answering the question of generating the nearest counterfactual explanation, it is essential to define the distance of the factual $x$ to the nearest counterfactual $\check{x}$. The literature has formed a consensus to use either the normalized $\ell_0$ or $\ell_1$ norm or any convex combination thereof (see for example [48, 39, 44, 23, 56, 59]). The $\ell_0$ norm puts a restriction on the number of feature changes between factual and counterfactual instance, while the $\ell_1$ norm restricts the average change:

$$c_0(\check{x}, x) = \frac{1}{d}\|x - \check{x}\|_0 = \frac{1}{d}\|\delta_x\|_0, \qquad c_1(\check{x}, x) = \frac{1}{d}\|x - \check{x}\|_1 = \frac{1}{d}\|\delta_x\|_1. \qquad (1)$$

**Constraint violation**   This measure counts the number of times the CE method violates user-defined constraints. Depending on the data set, we fixed a list of features which should not be changed by the used method (e.g., *sex*, *age* or *race*).

**yNN**   We use a measure that evaluates how much data support CEs have from positively classified instances. Ideally, CEs should be close to positively classified individuals which is a desideratum formulated by Laugel et al. [33]. We define the set of individuals who received an undesirable prediction under $f$ as $H^- := \{x \in \mathcal{D} : f(x) < \theta\}$. The counterfactual instances (instances for which the label was successfully changed) corresponding to the set $H^-$ are denoted by $\check{H}^-$. We use a measure that captures how differently neighborhood points around a counterfactual instance $\check{x}$ are

classified:

$$\text{yNN} = 1 - \frac{1}{nk} \sum_{i \in \breve{H}^-} \sum_{j \in \text{kNN}(\breve{x}_i)} |f_b(\breve{x}_i) - f_b(x_j)|, \qquad (2)$$

where kNN denotes the $k$-nearest neighbours of $x$, and $f_b(x) = \mathbb{I}[f(x) > 0.5]$ is the binarized classifier. Values of yNN close to 1 imply that the neighbourhoods around the counterfactual explanations consists of points with the same predicted label, indicating that the neighborhoods around these points have already been reached by positively classified instances. We use a value of $k := 5$, which ensures sufficient data support from the positive class.

**Redundancy**    We evaluate how many of the proposed feature changes were not necessary. This is a particularly important criterion for independence–based methods. We measure this by *successively* flipping one value of $\breve{x}$ after another back to $x$, and then we inspect whether the label flipped from 1 back to 0: e.g., we check whether flipping the value for the second dimension would change the counterfactual outcome 1 back to the predicted factual outcome of 0: $\mathbb{I}[f_b([\breve{x}_1, x_2, \breve{x}_3, \ldots, \breve{x}_d]) = 0]$. If the predicted outcome does not change, we increase the redundancy counter, concluding that a sparser counterfactual explanation could have been found. We iterate this process over all dimensions of the input vector.[3] A low number indicates few redundancies across counterfactual instances.

**Success Rate**    Some generated counterfactual explanations do not alter the predicted label of the instance as anticipated. To keep track how often the generated CE does hold its promise, the success rate shows the fraction of respective models' correctly determined counterfactuals.

**Average Time**    By measuring the average time a CE method needs to generate its result, we evaluate the effectiveness and feasibility for real–time prediction settings. We have included the run time measure to give users a rough estimate on what run times to expect when executing the respective algorithms.

## 5    Experimental Evaluation

Using `CARLA` we conduct extensive empirical evaluations to benchmark the presented counterfactual explanations methods using three real-world data sets. Our main findings are displayed in Figure 3, and Table 2. We split the benchmarking evaluation by CE method category. In the following Sections, we provide an overview over the used data sets (see Table 3) and the classification models. Detailed information on hyperparameter search for the CE methods is provided in Appendix E.

**Data sets**    The **Adult** data set [12] originates from the 1994 Census database, consisting of 14 attributes and 48,842 instances. The classification consists of deciding whether an individual has an income greater than 50,000 USD/year. Since several CE methods cannot handle non-binary categorical data, we binarized these features by partitioning them into the most frequent value, and its counterpart (e.g., *US* and *Non-US*, *Husband* and *Non-Husband*). The features *age*, *sex* and *race* are set as immutable. The **Give Me Some Credit** (GMC) data set [7] from a 2011 Kaggle Competition is a credit scoring data set, consisting of 150,000 observations and 11 features. The classification task consists of deciding whether an instance will experience financial distress within the next two years (*SeriousDlqin2yrs* is 1) or not. We dropped missing data, and set *age* as immutable.

| Data Set | Task | Positive Class | Size ($N \mid d$) | Features | Immutable Features |
|----------|------|----------------|-------------------|----------|--------------------|
| Adult | Predict Income | High Income (24%) | (45,222 \| 20) | Work, Education, Income | Sex, Age, Race |
| COMPAS | Predict Recidivism | No Recid. (65%) | (10,000 \| 8) | Crim. History, Jail & Prison Time | Sex, Race |
| GMC | Predict Financial Distress | No deficiency (93%) | (150,000 \| 11) | Pay. History, Balance, Loans | Age |

Table 3: Summarizing the used data sets, where $N$ and $d$ are the number of samples and input dimension after processing of the data. Results on the COMPAS data set are relegated to Appendix B.

**Black-box models**    We briefly describe how the black–box classifiers $f$ were trained. `CARLA` supports different ML libraries (e.g., Pytorch, Tensorflow) to estimate these classifiers as the implementations of the various explanation methods work particular ML libraries only. The first model is a multi-layer perceptron, consisting of three hidden layers with 18, 9 and 3 neurons, respectively. To allow a more extensive comparison (`AR` only works on linear models) between CE methods, we chose

---

[3]We do not consider all possible subsets of changes.

logistic regression models as the second classification model for which we evaluate the CE methods. Detailed information on the classifiers' training for each data set is provided in Appendix C.

**Benchmarking** For **independence based** methods, we find that no one single CE method outperformed all its competitors. This is not too surprising since algorithmic recourse is a multi–modal problem. Instead, we found that some methods dominated certain measures across all data sets. AR, AR-LIME, DICE performed strongest with respect to $\ell_0$ (see the top left panels in Figures 3a and 3b). AR-LIME does so despite our use of the LIME reduction. Therefore, it makes sense that AR, AR-LIME and DICE offer the lowest *redundancy* scores (Table 2a). CEM performed strongest with respect to the overall cost measure $\ell_1$ across data sets. GS is the clear winner when it comes to the measurement of time (Table 2a). Since the algorithm behind GS is based on a rather rudimentary sampling strategy, we expect that savvier sampling strategies should boost its cost performance significantly.

For **dependence based** methods, the results are mixed as well. While CLUE and REVISE are the winner with respect to the cost of recourse ($\ell_1$), the margins between these generative recourse models and the graph-based ones (FACE) are small (Figure 3). The FACE-EPS method performs strongest with respect to the $ynn$ measure (usually well above 0.60) (Table 2b) indicating that the generated CEs have sufficient data support from positively classified individuals relatively to the remaining dependence–based methods. As expected, the ynn measures are on average higher for the dependence based methods. This suggest that dependence based CEs are less often outliers. Notably, CLUE and REVISE perform best with respect to $\ell_1$ (with REVISE being the clear winner on 3 out of 4 cases), while they perform worst on $\ell_0$ – likely due to the decoder's imprecise reconstruction. In this respect, it is not surprising that these methods have average redundancy values that are up to twice as high as those by FACE. Finally, the generative model approaches (CEM-VAE, CLUE, REVISE) performed best with respect to time since the autoencoder training time amortizes with more samples.

# 6 Conclusion and Broader Impact of CARLA

The current implementations of recourse methods, mentioned in Section 4.1 are based on the original implementation of the respective research groups. Researchers mostly implement their experiments and models for specific ML frameworks and data sets. For example, some explanation methods are restricted to Tensorflow and are not applicable to Pytorch models. In the future, we will extend CARLA to decouple each recourse method from the frameworks and data contraints.

When trying to combine different CE methods into a common benchmarking framework we encountered the following issues: First, a great number of repositories only contain remarks about installation and script calls to recreate the results from the corresponding research papers. Second, missing information about interfaces for data sets or black–box models further complicated the process of integrating different CE methods into the benchmarking workflow. In order to add more CE methods and data sets to CARLA, we are currently in contact with several authors in this exciting and rapidly growing field. With a growing open-source community, CARLA can evolve to be the main library for generating counterfactual explanations and benchmarks for recourse methods. Therefore we are continuously expanding the catalog of explanation methods and data sets, and welcome researchers to add their own recourse methods to the library. To facilitate this process, we provide a step-by-step user-guide to integrate new CE methods into CARLA, which we present in Appendix A.

The rapidly growing number of available CE methods calls for standardized and efficient ways to assure the quality of a new technique in comparison with other approaches on different data sets. Quality assurance is a key aspect of actionable recourse, since complex models and CE mechanisms can have a considerable impact on personal lives. In this work, we presented CARLA, a versatile benchmarking platform for the standardized and transparent comparison of CE methods on different integrated data sets. In the explainability field, CARLA bears the potential to help researchers and practitioners alike to efficiently derive more realistic and use–case–driven recourse strategies and assure their quality through extensive comparative evaluations. We hope that this work contributes to further advances in explainability research.

**Acknowledgements**

We would like to thank the anonymous reviewers, and Sohini Upadhyay, Annabelle Redelmeier and Amir-Hossein Karimi for helpful comments and suggestions.

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
