# OpenReview forum: "CARLA: A Python Library to Benchmark Algorithmic Recourse and Counterfactual Explanation Algorithms"
_NeurIPS.cc/2021/Track/Datasets_and_Benchmarks/Round1 — NeurIPS 2021 Datasets and Benchmarks Track (Round 1)_

### Official Review · Reviewer_BewK · 2021-06-28
**Complete benchmarking framework that needs some revisions**

**Rating:** 7
**Confidence:** 2

**Strengths:**

The major strength of the paper is that it contributes to explainability research and helps ensure consistency and transparency of evaluation of CE methods, by offering a complete benchmarking framework to test existing methods and implement custom ones.
It is thus a useful contribution to the broad ML community and the growing literature on CE.



**Weaknesses:**

The experimental evaluation of the paper could be improved. The paper mainly looks at three datasets (Adult and GMC in the main text and COMPAS in the appendix) but having a wider variety of datasets would be good. Although the choice of datasets seems reasonable, the authors could justify the choice of these datasets in the literature (e.g., are these datasets the most commonly used to evaluate CE methods?). Furthermore, the black-box models part (lines 262-268) was unclear. Adding a more detailed description and justification of the choices made would be helpful.

See also the comments in the fields below.


**Additional Feedback:**

See the previous fields.

**Clarity:**

The paper is generally well written. Some specific parts that could be improved are (i) the definition of yNN which is hard to parse, (ii) a short description of COMPAS dataset which is missing and should be part of the datasets paragraph since it is mentioned in the main text, (iii) the black-box models part which is unclear as it is currently written (see above).


**Correctness:**

The paper needs to clarify  the following parts (which are not immediate or clear).
- Why is k=5 chosen in yNN? Why are not more parameters tested/ allowed?
- Regarding the definition of redundancy, the authors write in footnote 4 that they don’t consider all possible subsets of changes. How is this justified? There is potentially a dimensionality (and thus a computational efficiency) issue but how reliable can then this measure be (at least approximately)?



**Documentation:**

The datasets are well known and commonly used. The description of the datasets could be better and more detailed (this part could at least be added to the appendix if there are space constraints).

**Ethics:**

A brief discussion of general ethical concerns around the use of CE methods is missing and should be included.

**Relation To Prior Work:**

The paper offers an extensive overview of the CE and recourse literature and relevant methods. However, it is unclear whether this framework is the first of its kind for CE methods (although it seems to be).


**Summary And Contributions:**

The paper presents the open-source Python library CARLA  (Counterfactual And Recourse LibrAry), that provides a unified framework and competitive baselines to benchmark new counterfactual explanation (CE) and recourse methods across different datasets and ML models. More specifically, CARLA constitutes a benchmark of 11 existing counterfactual explanation methods, and provides a framework for comparison of future counterfactual explanation methods  based on a set of evaluation measures and data sets. CARLA also provides support for optimization frameworks (e.g., PyTorch, Tensorflow) and custom implementations.

The paper gives an overview of the mathematical definitions of counterfactual explanation and resource concepts and then proceeds by giving an overview of the methods implemented by CARLA. Then, they define the following evaluation measures: (i) costs, defined as the normalized $l_0$ or $l_1$ distance (or their convex combination) of the factual $x$ to the nearest counterfactual $\tilde{x}$, (ii) constraint violation, i.e., the number of times a CE method violates user constraints (e.g., the sex of a person should be immutable), (iii) yNN, which uses a k-nearest neighbors (kNN) to reflect how differently neighborhood points around a counterfactual instance $\tilde{x}$ are classified (ideally this measure is close to 1), (iii) redundancy, which measures the number of unnecessary changes in the features, (iv) average time, (v) success rate.

For the evaluation, the authors use three datasets, namely Adult, Give Me Some Credit (GMC), and COMPAS. For both independence- and dependence- based methods, the paper finds that no single CE method was a clear winner, although certain methods outperformed the rest with respect to a subset of measures as the paper describes in greater detail.

---

> ### Author Response · Authors · 2021-07-10
> **Thank you for your valuable comments and suggestions.**
>
> Justification for the choice of data sets
>
> - We have chosen these data sets for two reasons: first, these data sets originate from domains where counterfactual explainability could play a useful role: e.g., the GiveMeCredit data set considers whether customers, who apply for loans, will default on these loans or not. While black-box algorithmic methods are increasingly being deployed in real-life consequential situations such as credit decisions, we have selected the data sets to provide a realistic testbed on how counterfactual explanation methods could be applied in practical use-cases.
> Second, these data sets have also been used in many of the prior works on counterfactual explanations (see for example Ustun et al (2019), Karimi et al (2020) and Pawelczyk et al (2020), among others).
>
> Justification for the choice of classifiers
>
> - Most methods providing counterfactual explanations were developed for gradient-based classifiers (e.g., Wachter, REVISE, CRUDS, CEM, etc.). More recently, explanation methods have been proposed that also work with decision tree-based methods (e.g., FOCUS). We are currently working on providing support for finding counterfactual explanations for tree-based classifiers (e.g., Feature Tweaking, FOCUS) as well, and will we add this functionality to our library as soon as it is available.
>
> Measures
> - ynn and k=5: Choosing very low values such as 1 or 2 would not provide a meanigful measure of whether a counterfactual explanation is a potential outlier or not. On the other hand, it is a great suggestion to experiment with ynn as function of k for sufficiently large k. We will add these results to the appendix.
>
> - Redundancy: This measure is more important for the independence based methods as these methods assume that feature changes can be made regardless of feature correlations. Essentially, this assumption is also made when computing the redudancy measure. As a consequence, for the dependence based methods this measure is less important. We will add these considerations to the manuscript.

---

### Official Review · Reviewer_yc5p · 2021-07-05
**The paper presents a benchmarking framework to consistently evaluate counterfactual explanation methods. The contribution, is timely and, in general, well-implemented by the authors.**

**Rating:** 8
**Confidence:** 5

**Strengths:**

As the authors mention in the introduction, algorithmic recourse is a nascent field gathering a lot of interest from researchers in the ML community. There are numerous technical methods proposed in the last couple of years, however, there is no common framework/measure to evaluate their effectiveness. In that sense, the paper is very strong because of its timeliness.

Other than that, I found the choice of evaluation measures reasonable since they capture most of the main aspects of problems related to algorithmic recourse. I think this library has the potential to be widely used by researchers in the area.

**Weaknesses:**

As the authors mention in Section 2 (line 117), their library does not support counterfactual explanation methods that are based on structural causal models. However, such models are increasingly appearing in the related literature [1,2,3] and they may allow the study of different aspects of algorithmic recourse, e.g., behavioral aspects of counterfactual explanations [4,5,6]. Even though this is not a major drawback of their library, I would encourage the authors to explore additional ways of supporting methods based on causal models and to consider additional metrics able to measure, for example, the fairness or the ability of manipulation of algorithmic recourse methods.

[1] https://arxiv.org/abs/2002.06278
[2] https://arxiv.org/abs/2006.06831
[3] https://arxiv.org/abs/2010.06529
[4] https://arxiv.org/abs/2106.02666
[5] https://arxiv.org/abs/2002.04333
[6] https://arxiv.org/abs/1910.10362

--- After author response ---

I read the response and I would like to keep my current score. I encourage the authors to keep working actively on their library to integrate new methods for algorithmic recourse and counterfactual explanations as they gain prominence in the literature.

**Additional Feedback:**

For suggestions for improvement, look at the Weaknesses section.

**Clarity:**

The paper was well structured and easy to follow. Two minor things:
Line 70: “can” is written twice.
Line 249: The authors mention three datasets. However, it would be best to clarify that the 3rd one is in the appendix.

**Correctness:**

As I mentioned in “Strengths”, the evaluation methods are well-thought by the authors and consistent with the existing methods in the related literature.

**Documentation:**

The authors present sufficient detail about the implementation of the main parts of their library and, a discussion about the technical details and the choice of hyperparameters can be found in the appendix. Moreover, the source code is publicly available on GitHub and it is well documented, allowing researchers in the area to contribute to it.

**Ethics:**

Does not apply.

**Relation To Prior Work:**

As mentioned earlier, there is no common benchmarking framework for algorithmic recourse methods. In that sense, this work is novel. Perhaps, I would have liked a discussion of other evaluation criteria appearing in the literature since some of the metrics used by the authors have probably appeared as evaluation measures (perhaps in slightly different forms) in prior work.

**Summary And Contributions:**

The authors present CARLA, a library to benchmark methods of counterfactual explanations/algorithmic recourse. First, they provide an overview of the main techniques/approaches of current methods. Then, they briefly explain 11 methods which they evaluate in the paper and they propose 6 evaluation measures of interest. Lastly, using 2 datasets common in the related literature, they benchmark the considered methods using the evaluation measures they proposed and they briefly discuss the quantitative results.

---

> ### Author Response · Authors · 2021-07-10
> **Thank you for the valuable comments and suggestions.**
>
> On Support for Causal Methods
>
> - We very much agree with you that it would be great to support causal recourse models as these types of models become increasingly researched. Currently, we're extending our library for tree based classifiers and support for higher cardinality categorial features. While we already have working implementations of some of your suggested papers (e.g., [1]), we have not started integrating these into our library, yet. We will certainly do this in the future.

---

### Official Review · Reviewer_RLpD · 2021-07-05
**CARLA is a python-library implementing popular counterfactual explanation algorithms. It allows easy benchmarking of new algorithms. The paper's contributions are important but limited.**

**Rating:** 7
**Confidence:** 3
**Correctness:** 1. See my comment about 'average time…
**Clarity:** Yes, the paper was easy to follow.

**Strengths:**

The paper introduces a python library, CARLA, to easily benchmark new approaches in this growing subarea of Explainability. This library is of interest to the counterfactual explanation research community, which can use CARLA to compare new algorithms against benchmarks. Further, this library should be of interest to the broader research community and practitioners, who can utilize this to test and pick an appropriate algorithm for their application.

However, we note that the authors use pre-existing methods and datasets. Thus, their contribution is limited to creating a library that implements existing algorithms and makes it easier for researchers to benchmark new algorithms.

The authors compare the popular counterfactual explanation algorithms on several natural parameters computed on standard datasets. Their empirical methodology appears to be sound (also see Weaknesses).

**Weaknesses:**

I have two suggestions/concerns, which are largely fixable.

1. Currently, the benchmarks do not consider categorical attributes (none of the algorithms reported in Table 1 handle categorical attributes). I think reporting results on non-binary attributes is important because non-binary attributes frequently arise in practice. Some works on counterfactual explanations handle categorical attributes (e.g., [[1907.02584]](https://arxiv.org/pdf/1907.02584.pdf)), and it would be good to include them in the benchmarks.

2. As the authors mention, different counterfactual explanation methods are implemented in different machine learning libraries. As a result, some methods could be slower than due to the libraries-specific optimizations they use (and not due to the inherent differences in the methods). Is 'average time' a useful measure to compare the efficiency of two algorithms implemented using different libraries? If not, is there a better measure of the efficiency of different methods?

- I understand that the value of 'average time' could be useful to give a rough estimate of the running time. However, if this is the main goal, then it can be useful to clearly state this in the paper and Github repo.

--- After the author response ---
I thank the authors for confirming my comment on the "average time" measure. and clarifying that they added two methods supporting nonbinary protected attributes to the library. I hope the authors would add some discussion on these in the final version. I am increasing my rating.

**Additional Feedback:**

"We do not consider all possible subsets of changes": One could take a small number k as input (from the user), and consider all possible subsets that change at most k parameters at a time.

Stylistic suggestions:
In Figure 2, the edges could be made thicker. Currently, they are too thin to view the colors (especially in print).

Some typos:
1. Citation [39] seems to be a repeat of citation [40]
2. Citation [56] seems to be a repeat of citation [57]
3. Citation [61] seems to be a repeat of citation [60]
4. ‘ArXiv’ -> ‘arXiv’ in Figure 1
5. 'In this Section' -> 'In this section' (In the first line of several sections)
6. 'Let D be data sets of' -> 'Let D be a data set of' in line 134
7. 'Sections' -> 'sections' in line 250
8. 'overview over' -> 'overview of' in line 251


**Documentation:**

Yes, the supplementary material and the Github repository provide sufficient details for reproducibility.

**Ethics:**

No.

**Relation To Prior Work:**

The paper suitably reviews the prior work on counterfactual explanation algorithms.

It says that there is an "absence of widely available open-source [counterfactual explanation algorithm] implementations." Thus, there is no prior work directly related to this work. That said, I am not familiar enough with the literature to verify this claim.



**Summary And Contributions:**

Counterfactual explanation algorithms are a specific class of explanation algorithms that suggest a link between the output of ML algorithms and changes in their inputs. Counterfactual explanations is a growing field with numerous existing methods.

Any practitioner looking to deploy counterfactual explanation algorithms requires a systematic method to compare the existing methods and choose an appropriate method for their context. Further, systematic benchmarks are also crucial to ensure high-quality research in this area.

The authors develop an open-source python library, CARLA, that allows users to easily benchmark compare different counterfactual explanations algorithms against each other on standard datasets. Currently, CARLA has a modest number (11) of counterfactual explanation algorithms and gives users the option to include their custom algorithms.

---

> ### Author Response · Authors · 2021-07-10
> **Thank you for the valuable comments and suggestions.**
>
> Average time
>
> - We implemented the respective methods faithfully, following either the original pseudo code suggestions or the original github implementations. We have included the run time measure to give users an idea on what run times to expect when executing the respective algorithms.  You are right that the run times can vary depending on implementation choices such as the chosen library (e.g., Tensorflow vs. PyTorch vs. Sklearn). As such the reported run times provide rough estimates. We will add these clarifications to our manuscript.
>
> Currently no support for high cardinality categorical attributes
>
> - We have not considered categorical attributes with more than 2 categories, yet. The reason is the following: most of the suggested methods (e.g., FACE, AR, Wachter, GS, etc.) have not considered this aspect in their works. To allow a comparison among a larger number of competing methods, we currently only consider a baseline comparison where we allow for 2 categories per feature. We think that this is the least common denominator providing common ground for the benchmarking process.
>
> - That being said, ever since we submitted our work here, we have added two additional methods which can handle higher cardinality categorical features (e.g., C-CHVAE,  CRUDS) and one more will follow soon (the work you pointed to). We are currently also working on providing support for finding counterfactual explanations for tree-based classifiers. We will be continuously updating our benchmark, and to see what features we currently support we invite you to have a look at our documentation: https://carla-counterfactual-and-recourse-library.readthedocs.io/en/latest/. The documentation also details how users can contribute new algorithms and data sets to our package.

---

### Decision · Program_Chairs · 2021-07-26

**Decision:**

Accept

**Comment:**

While the library reuses some existing datasets, reviewers agree that this library is useful as a unified framework for counterfactual explanations and recourse methods. 11 existing methods are benchmarked across three datasets, and the package supports both pytorch and tensorflow. Evaluation metrics are reasonable. A few questions raised by reviewers are promptly answered by authors. Authors also are looking into expanding this work to CSMs.